 **eLIFE**

# Sequential conformational rearrangements in flavivirus membrane fusion

**Luke H Chao[1,2], Daryl E Klein[1,2], Aaron G Schmidt[1,2], Jennifer M Peña[1,2], Stephen C Harrison[1,2,3]\***

[1]Department of Biological Chemistry and Molecular Pharmacology, Harvard Medical School, Boston, United States; [2]Laboratory of Molecular Medicine, Boston Children's Hospital, Boston, United States; [3]Howard Hughes Medical Institute, Harvard Medical School, Boston, United States

**Abstract** The West Nile Virus (WNV) envelope protein, E, promotes membrane fusion during viral cell entry by undergoing a low-pH triggered conformational reorganization. We have examined the mechanism of WNV fusion and sought evidence for potential intermediates during the conformational transition by following hemifusion of WNV virus-like particles (VLPs) in a single particle format. We have introduced specific mutations into E, to relate their influence on fusion kinetics to structural features of the protein. At the level of individual E subunits, trimer formation and membrane engagement of the threefold clustered fusion loops are rate-limiting. Hemifusion requires at least two adjacent trimers. Simulation of the kinetics indicates that availability of competent monomers within the contact zone between virus and target membrane makes trimerization a bottleneck in hemifusion. We discuss the implications of the model we have derived for mechanisms of membrane fusion in other contexts.

## Introduction

Cell entry of enveloped viruses requires membrane fusion, catalyzed by a viral surface protein. The fusion protein of flaviviruses—the group that includes yellow fever (YFV), West Nile (WNV), dengue (DV), and tick-borne encephalitis (TBEV) viruses—is the envelope protein (E), which becomes fusogenic when exposed to reduced pH in an endosome (*Allison et al., 1995*; *van der Schaar et al., 2007*). This step merges viral and endosomal membranes and releases the viral genome into the cytosol.

Although thermodynamically favorable, lipid-bilayer fusion has a very high kinetic barrier (~50 kcal/mol) (*Rand and Parsegian, 1984*). Modeling and experiment suggest that hydration-force repulsion occurs when two bilayers approach more closely than 15–20 Å. Fusion proteins such as E reduce this barrier by inducing the distortion needed to form a hemifusion stalk—the short-lived intermediate that resolves to form a pore. They do so through a series of membrane-coupled conformational rearrangements (*Harrison, 2008*).

Mature flavivirus particles have an ordered icosahedral lattice of E dimers on the surface of a ~500 Å diameter virion (*Figure 1A*) (*Zhang et al., 2013a*). Cleavage by furin of prM, the E-protein 'chaperone', during transit through the trans-Golgi network converts an immature, non-infectious, and fusion incompetent particle, with trimer-clustered prM-E spikes, into an infectious and fusion competent particle, with dimer-clustered E (*Rey et al., 1995*; *Stadler et al., 1997*; *Li et al., 2008*). This rearrangement primes the E protein to undergo a sequence of fusion-inducing conformational changes when exposed to acidic pH.

Flavivirus E proteins have a conserved, three-domain architecture, with a C-terminal connector ('stem') linking the third domain to the transmembrane anchor (*Rey et al., 1995*). Domain I is a central

**\*For correspondence:** harrison@crystal.harvard.edu

**eLife digest** Flaviviruses are a group of viruses that cause serious diseases in humans, including yellow fever, West Nile fever and dengue fever. Like all viruses, flaviviruses protect their genetic material with a protein shell and, like many other viruses, that shell also has a lipid membrane.

Flaviruses use one of their surface membrane proteins, known as 'envelope protein' or simply 'E', to bind to the surface of host cells. Once the virus has attached to the host cell membrane, it becomes engulfed within a bubble-like structure called an endosome, which also has a surrounding membrane. The interior of an endosome is acidic. Under these conditions the E protein undergoes a series of changes that bring the two membranes into close contact, so that the membrane of the virus can fuse with the membrane of the endosome. This membrane fusion allows the genome of the virus to escape the endosome and hijack the cell to make new copies of the virus.

The E proteins on a mature flavivirus particle are found in pairs, but previous work showed that these proteins must work together in groups of three (called 'trimers') for the viral and endosomal membranes to fuse. Chao et al. have now asked: what are the rate-limiting steps that lead to the formation of trimers? And how many trimers are necessary to cause the membranes to fuse?

Chao et al. have investigated these questions using virus-like particles containing the E protein of West Nile Virus. They used techniques that can track individual particles, which their laboratory had previously used to investigate the influenza virus, to model changes in the E protein before, during and after membrane fusion. Chao et al. then made mutant versions of the envelope protein and used virus-like particles containing them to test the model.

The data that Chao et al. obtained and computer simulations they carried out suggest that exposure to acidic conditions encourages the pairs of E proteins to separate and extend towards the endosome membrane. Individual E proteins then group together into trimers, and at least two trimers are needed to exert enough force to allow the membranes to fuse. The experimental design used by Chao et al. will now allow them to study the action of molecules that inhibit membrane fusion by West Nile Virus and other viruses.

β-barrel that positions the other two domains; domain II, two long, clustered extensions from domain I, bears the fusion loop at its distal tip; domain III is an immunoglobulin-like domain that may have a receptor-binding surface. In the prefusion, neutral-pH conformation, a cavity at the junction between domains I and III sequesters the fusion loop of the E dimer partner (*Figure 1B*). Protonation of key histidine residues induces dimer dissociation, exposing the fusion loop and allowing the ectodomain to hinge outwards from the virion surface (*Fritz et al., 2008*). Engagement of the fusion loop with the target membrane facilitates trimerization, accompanied by repositioning of domain III relative to domain I (*Allison et al., 1995*; *Rey et al., 1995*; *Modis et al., 2004*). Rearrangement of the membrane-proximal stem, from a conformation buried in the outer leaflet of the viral membrane to one that stabilizes the inter-protomer trimer interface, can then 'zip' the trimer together and induce collapse of the extended intermediate, causing the transmembrane anchors to approach the trimer-clustered fusion loops (*Pangerl et al., 2011*; *Klein et al., 2013*). This last step favors formation of a hemifusion stalk between the apposed leaflets of the two membranes. Disruption of domain III repositioning by exogenous free domain III or inhibition of stem zippering by stem-derived peptides inhibits infection (*Liao and Kielian, 2005*; *Schmidt et al., 2010*). Resolution of the hemifused state into an open pore may require further rearrangement of the fusion loops and the C-terminal transmembrane anchors.

We have studied the kinetics of WNV membrane fusion, in a single-particle format developed previously to analyze influenza virus fusion (*Floyd et al., 2008*) (*Figure 1C*). Kinetic data can define rate-limiting steps preceding an observed process, and coupled with site-directed mutation they can identify key intermediates (*Floyd et al., 2010*; *Ivanovic et al., 2013*). We have used this approach to probe the conformational transition of WNV E on the virion surface. Our data support a mechanism in which a pH-dependent dimer-monomer equilibrium creates, at the contact between virus and target membrane, a pool of monomers competent to cluster into target-membrane engaged trimers. Formation of at least two adjacent trimers allows progression to hemifusion. The corresponding two-stage structural picture is (i) that stochastic activation of E monomers (by dimer dissociation and monomer outward extension) leads to trimerization and essentially irreversible target-membrane

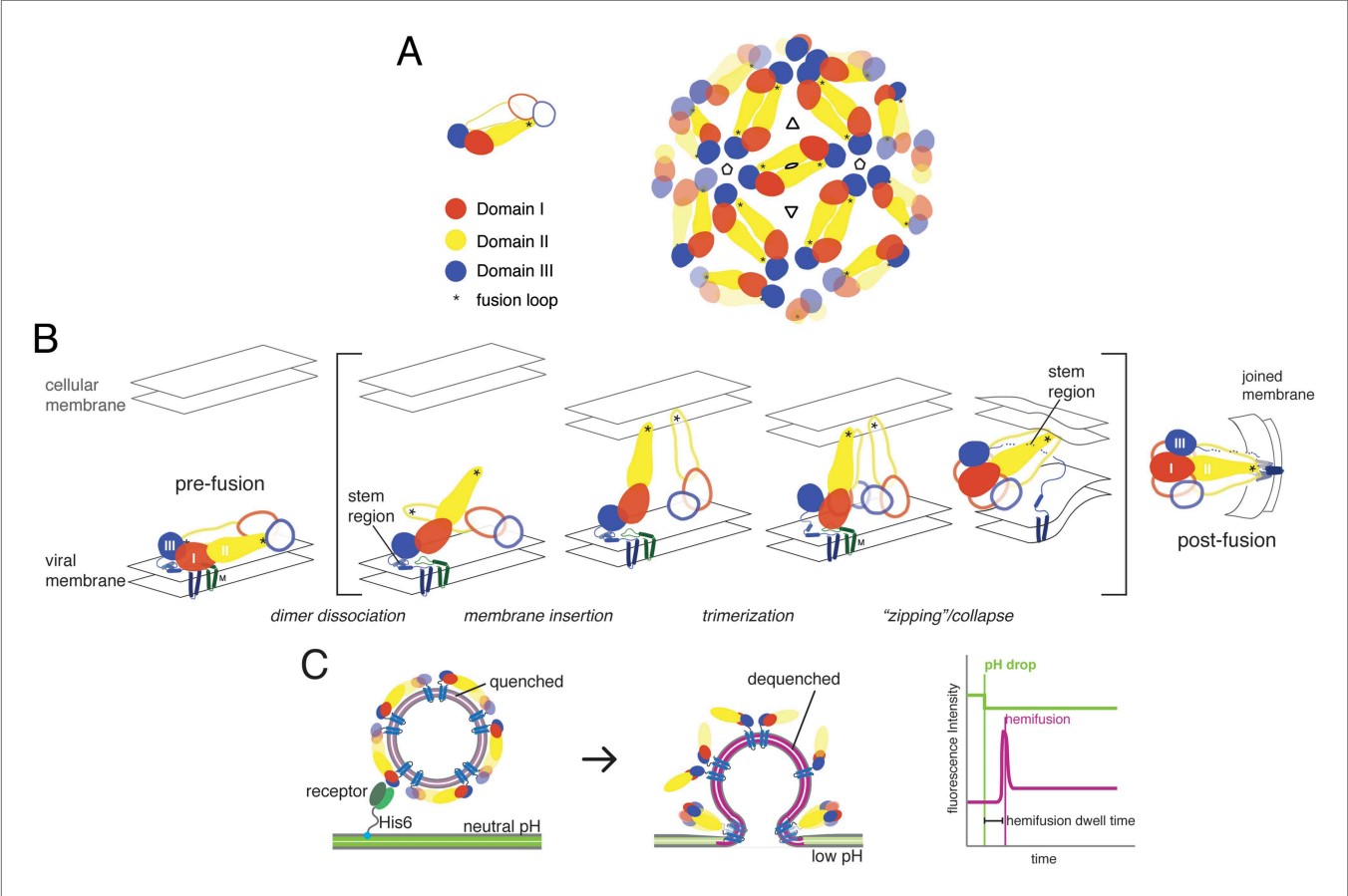

**Figure 1**. Molecular structure and conformational rearrangements in flavivirus membrane fusion. (**A**) Left: domains of flavivirus E protein in the dimer on the surface of a mature virion or virus-like particle (VLP), monomers shown in solid and outline forms. right: arrangement of dimers on a small (60-subunit) VLP. (**B**) Schematic diagram of the low-pH induced, fusogenic transition in E, with postulated intermediates enclosed in brackets. The steps illustrated are: dissociation of the dimer and outward projection of each monomer, membrane engagement of the fusion loops (asterisks), trimerization of membrane-interacting monomers, collapse of trimer into the postfusion conformation. (**C**) Schematic diagram of the single-particle fusion assay used in these studies. Left: DiD-labeled VLPs are captured on the supported bilayer at neutral pH through a surrogate receptor (Fab) anchored to $Ni^{2+}$/NTA-headgroup lipids; center: low-pH induced fusion and DiD dequenching; right: idealized plots of DiD fluorescence within the diffraction limited zone of the captured VLP, in which lateral diffusion and loss of intensity follows an initial burst when dequenching reports merger of proximal leaflets of the two membranes; a pH-dependent fluorophore in the membrane reports the time of pH drop.

engagement, whenever three neighboring monomers are active, and (ii) that two adjacent trimers can together exert enough force on the membranes they bridge (target membrane and viral membrane) to produce a hemifusion stalk. Formation of adjacent trimers is limited by the availability of competent monomers. This mechanism—like the one shown previously to fit influenza-virus fusion kinetics—does not require defined trimer–trimer interactions, because resistance of the two membranes to deformation toward a hemifusion stalk couples conformational changes in the trimers that bridge them (*Ivanovic et al., 2013*). Fusion proceeds rapidly whenever a sufficient number of them can overcome the deformation barrier. We suggest that this description may apply more generally to fusion of intracellular vesicles and to fusion of two cells.

## Results

### Single-particle analysis of WNV hemifusion

Recombinant expression of flavivirus proteins prM and E yields mature, non-infectious, empty virus-like particles (VLPs). The properties of these particles are essentially the same as those of virions (*Figure 2A–C*) (*Schalich et al., 1996*). WNV VLPs were prepared as described in Methods. Electron

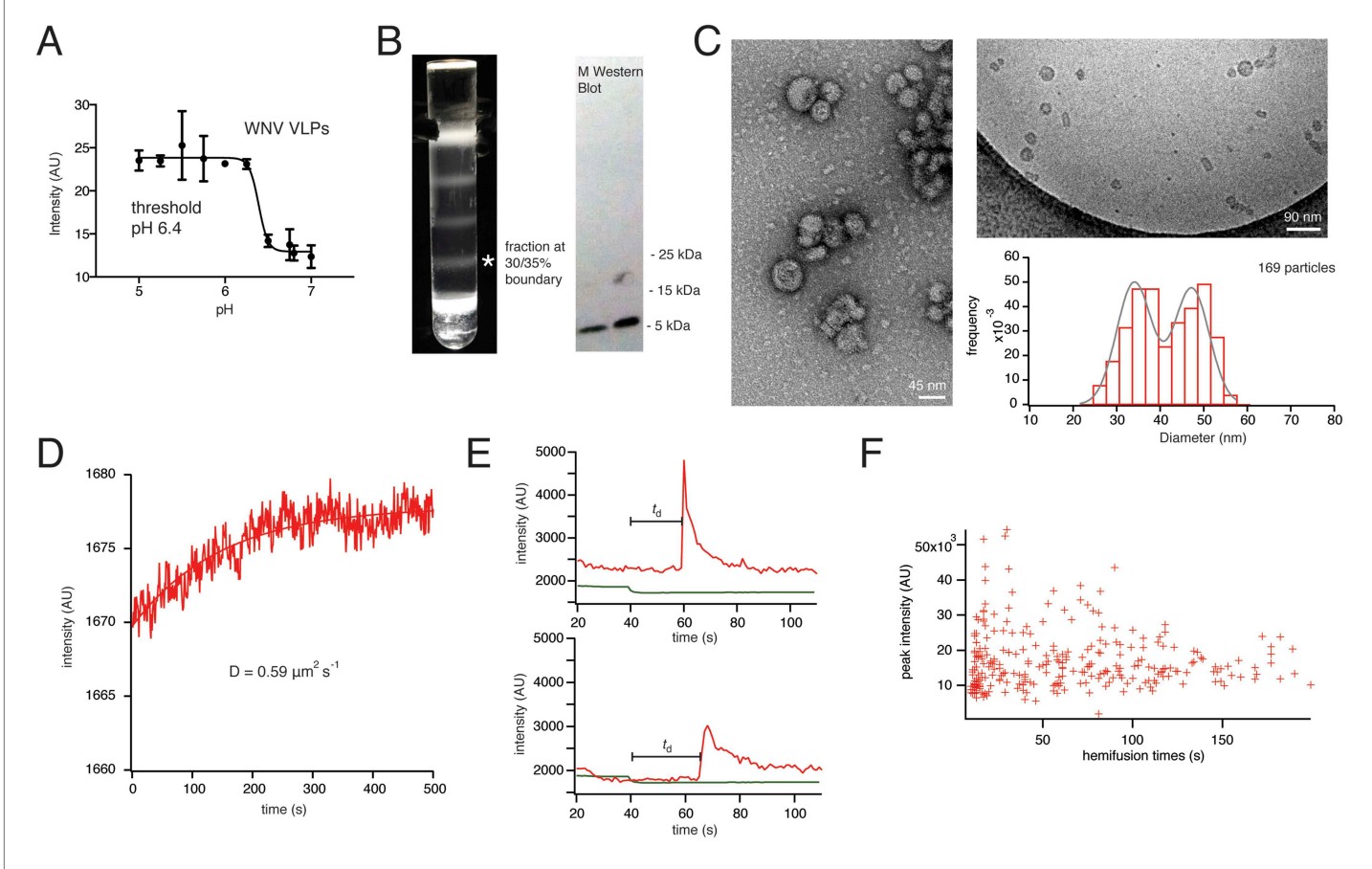

**Figure 2**. Characterization of VLPs and single-particle fusion. (**A**) Bulk liposome fusion activity of purified WNV VLPs as function of pH, measured by DiD dequenching upon fusion with liposomes. (**B**) Discontinuous Optiprep gradient of WNV VLPs. Immunoblot at right (left lane, 25/30% fraction; right lane, 30/35% fraction), showing >95% cleavage of prM to M. (**C**) Negative-stain (left) and cryo- (right) electron microscopy of WNV VLPs, with histogram of size distribution of cryo-electron micrograph particles. The smaller size corresponds to 60-subunit VLPs; the larger, to 180-subunit, virion-size VLPs. (**D**) Fluorescence recovery after photobleaching for a region of supported bilayer, showing recovery with a diffusion constant D = 0.59 μm²/s, consistent with a fluid bilayer, calculated from D = 0.22 R²/t (**Takamori et al., 2006**), where R is radius of photobleached area. (**E**) Representative single-particle traces for fluorescein bleaching upon pH decrease (green) and DiD dequenching and diffusion (red). The bar shows the lag time (or 'dwell time', $t_d$) between pH drop and time of half-maximal dequenching. (**F**) The peak intensity of the hemifused particles does not correlate with the hemifusion lag time.

microscopy shows particles in two principal size classes, corresponding to the 60- and 180-E subunit shells described for TBE VLPs (**Figure 2C**) (**Allison et al., 2003**). We measured a bulk liposome hemifusion pH threshold of ~6.4 for WNV VLPs, as previously reported (**Moesker et al., 2010**) (**Figure 2A**).

We collected single-particle data for WNV VLP hemifusion by total internal reflection fluorescence (TIRF) microscopy to obtain single-particle histograms of hemifusion delay times (the interval between fluorescein signal decrease and DiD dequenching) for VLPs (**Video 1**). The VLP or virus preparation was introduced at neutral pH, and the pH then lowered to induce fusion. To uncouple attachment from fusion-loop exposure, we incorporated either the E16 Fab or the lectin domain of DC-SIGN-R as a receptor (**Davis et al., 2006**), with a C-terminal His6 tag that bound a Ni-NTA headgroup lipid in the fluid (**Figure 2D**), glass-supported, lipid bilayer (**Figure 1C**). In these experiments, loss of fluorescence from a fluorescein-conjugated lipid in the bilayer reported the pH drop, and dequenching of a hydrophobic fluorophore (DiD) introduced into the particle membrane reported the time of membrane merger and dye diffusion into the bilayer (**Figure 2E**).

The extent of labeling did not affect the course of VLP hemifusion, as shown by the lack of correlation between peak intensity and hemifusion time (**Figure 2F**). We collected WNV VLP single particle data over a pH range from 5.0 to 6.25 (**Figure 3A**). We also collected data for live-virus Kunjin, a

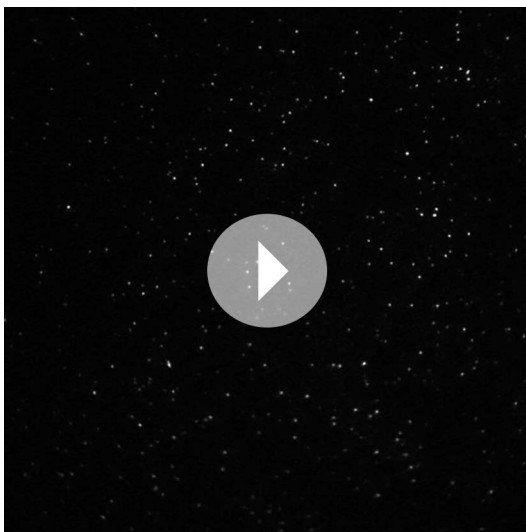

**Video 1**. Video of WNV VLP hemifusion events at pH 6.25, recorded at 640 nm channel sped up 20 times actual speed.

variant of West Nile, and found similar kinetics over the pH range from 5.0 to 6.0 (*Figure 3B*). Labeling of the Kunjin virus with DiD did not influence infectivity (*Figure 3C*). The absence of correlation between intensity of DiD fluorescence and VLP hemifusion times and the agreement between data for WNV VLPs and intact Kunjin virus show that hemifusion kinetics do not depend on particle size (i.e., on whether they are 60- or 180-subunit particles) (*Figures 2F and 3B*).

We compared bulk hemifusion rates over a range of temperatures, 20–36°C, at pH 6.25, just below the pH threshold, and found an Arrhenius dependence for the time to hemifusion (*Figure 4A*). The absence of abrupt changes in rate allows us to conclude that the protein conformational changes relevant for inducing hemifusion are largely the same across this temperature interval and that there are no detectable effects of lipid phase transitions. We therefore chose to work at 20°C for single-particle data collection, to slow the reaction and thereby facilitate detection of transient intermediates.

Production temperature can affect the stability of Dengue E packing on the virion surface and alter epitope accessibility of E in WNV (*Zhang et al., 2013b*). We tested WNV VLPs produced at 37°C and 28°C and found no difference in the extent of prM processing or in the bulk liposome fusion activity. Dengue virus serotype 4 VLPs were significantly less active when produced at 37°C (*Figure 4B*), consistent with previous observations (*Zhang et al., 2013b*).

The mean delay time in the WNV VLP single particle distributions decreased with increasing proton concentration (*Figure 3A*). We could fit the data at lower pH (pH 5.0–5.5) with a probability density function describing a single exponential decay and the data at higher pH (pH 5.75–6.25) with a function describing a process with two steps of equal rate, either parallel or sequential (see 'Materials and methods'). These qualitative conclusions were independent of the surrogate receptor, as data collected with DC-SIGN-R had only slightly shorter delay times and slightly higher hemifusion rates (*Figure 4C*). We observed from single-particle data for Kunjin-virus fusion between pH 5.0 and 6.0 mean delay times similar to those we measured for WNV VLPs, with exponential fits to a formal rate constant for the lower pH range and two-step fits at pH 5.75 and above (*Figure 3B*).

WNV VLPs that incorporated a fusion-loop mutant (W101A) did not dequench upon pH drop and instead released and washed away. This result is consistent with a requirement for fusion-loop insertion into the supported bilayer, as assumed in models for the reaction. It also shows that attachment and fusion-loop interaction are distinct steps in these experiments. In bulk fusion studies, in which acidification induces fusion with liposomes that do not bear a receptor or surrogate, exposure of the fusion loops may be necessary even for initial interaction of the virus or VLPs with the target bilayer. Mixed VLPs, with a 50:50 mixture of wild-type:W101A E protein, had a mean dwell time (52–53 s at pH 5.5) and an overall yield of fusion events (~30%) similar to those of fully wild-type particles (*Figure 4D*).

## An initial reversible pH-dependent transition

The first step in assembly of trimeric fusion complexes is release of E protein ectodomains from lateral contacts, especially those between dimers, in the surface of a mature virion. Some epitopes inaccessible in the pre-fusion surface lattice as visualized by cryoEM (*Mukhopadhyay et al., 2003*) bind antibodies even at neutral pH, showing that E subunits transiently expose buried surfaces (*Dowd et al., 2011*). The temperature dependence for expansion of dengue virions suggests that thermal fluctuations can potentiate access to such cryptic epitopes (*Zhang et al., 2013b*; *Zhang et al., 2014*). The antibody-captured and temperature-induced states are probably on-pathway intermediates to subsequent hemifusion and pore formation.

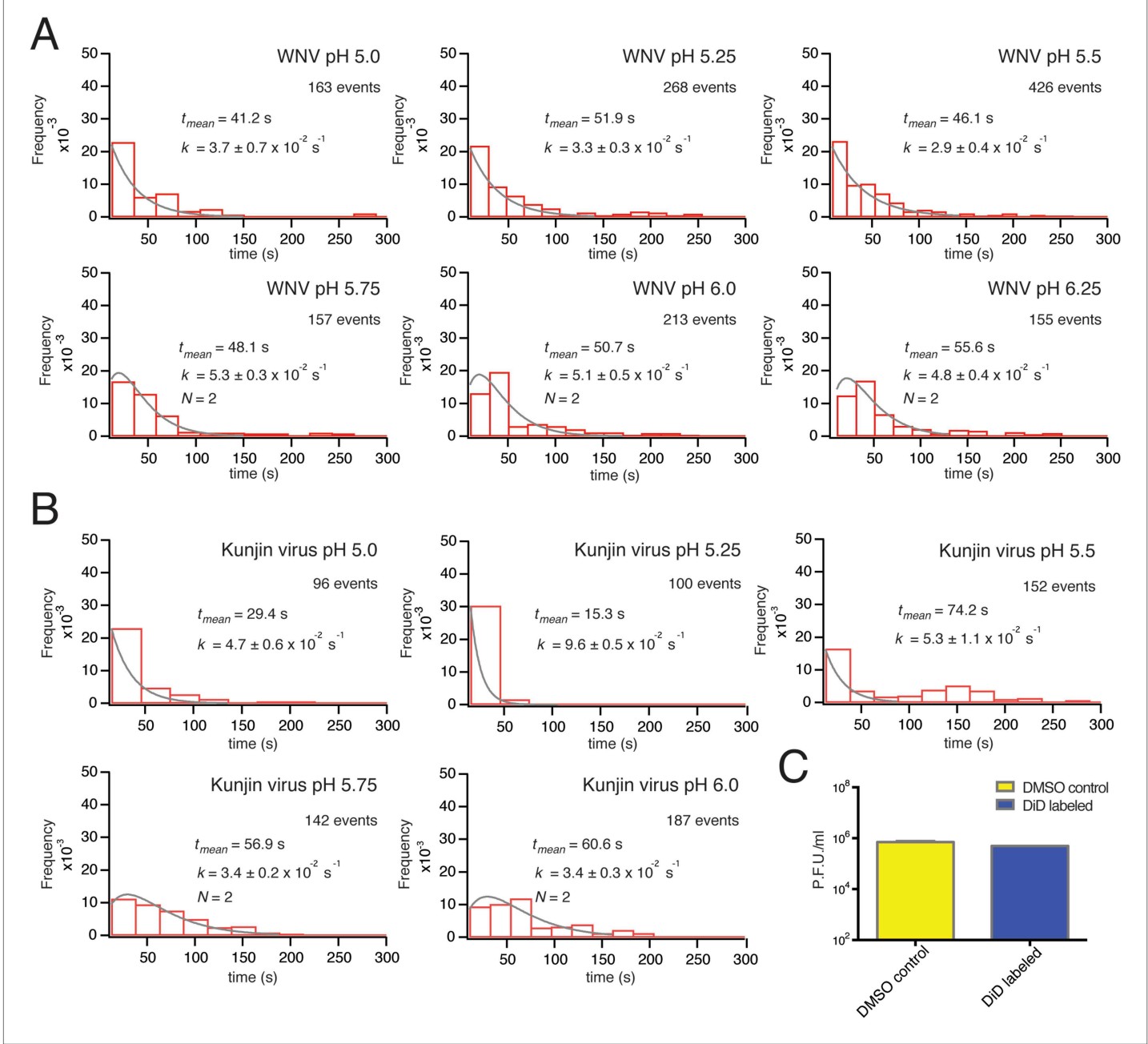

**Figure 3**. Single-particle WNV VLP and Kunjin virus hemifusion. (**A**) Histograms of single-particle dwell times for WNV VLPs, pH 5.0 to 6.25, with fitted curves calculated for a process with a single exponential decay (pH 5.0–5.5) or a gamma distribution with two sequential or parallel steps ($N = 2$, pH 5.75–6.25) (fits calculated over the first 150 s of the data). The number of events detected and the mean dwell time ($t_{mean}$) are noted (mean dwell time determined from all data). (**B**) Histograms of single-particle dwell times of Kunjin virus particles, with fits to a single exponential decay (pH 5.0–5.5) or a gamma distribution with two sequential or parallel steps ($N = 2$, pH 5.75–6.25). (**C**) Comparison of infectivity of carrier (DMSO) and DiD-treated Kunjin virus.

We measured by dynamic light scattering the hydrodynamic radius, R, of Kunjin virus as a function of pH, to detect initial low pH-induced rearrangement of E proteins on the virion surface (***Figure 5A***). The uniform size of the virus particles (as opposed to VLPs) allowed us to interpret an increased R as projection of monomeric E subunits away from the particle surface, exposing the fusion loops. R increased reversibly as we lowered the pH, with a transition midpoint at pH 6.8 (***Figure 5B***) The value of R at high pH is larger than the outer radius of the fully ordered, compact cryoEM structure,

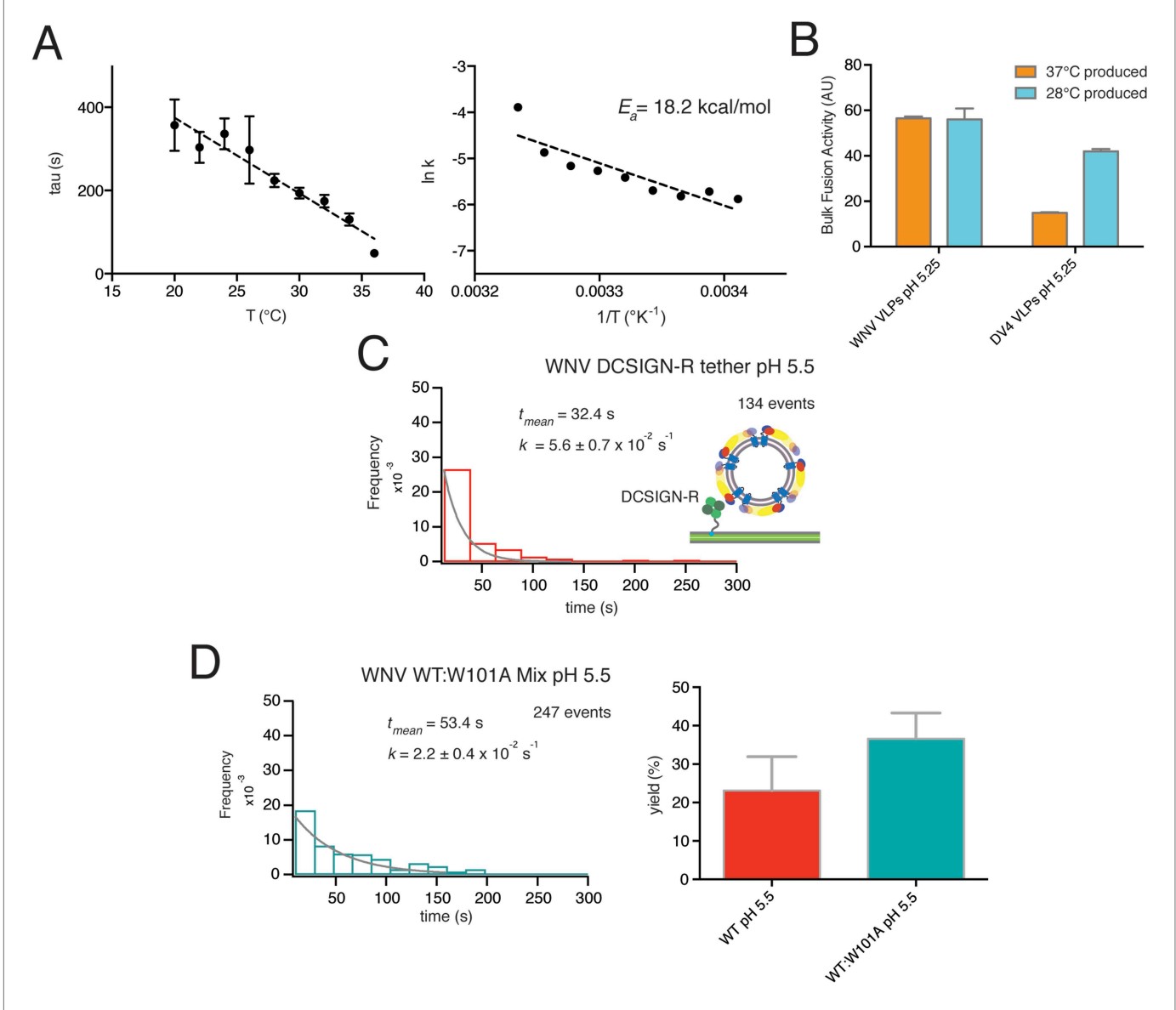

**Figure 4**. Data for temperature dependence, receptor dependence, and sensitivity to fusion-loop mutation for fusion of WNV VLPs. (**A**) Kinetics of bulk hemifusion for WNV VLPs. Left: time to hemifusion as a function of temperature between 20°C and 36°C; right: Arrhenius plot and calculated activation energy. (**B**) Overall bulk fusion activity for WNV and Dengue serotype 4 (DV4) VLPs produced at 28°C and 37°C. (**C**) Single-particle kinetics of WNV VLP fusion with DCSIGN-R as receptor at pH 5.5; compare with *Figure 3A*, upper right-hand panel. (**D**) Single-particle fusion kinetics (left) and fusion yield (right) for WNV VLPs containing a 1:2 mixture of mutant (W101A) and wild-type E; compare with *Figure 7D*.

suggesting that at 20°C and pH 8 there is enough outward 'breathing' to influence hydrodynamic drag, but some of the effect could be due to residual polydispersity. The Hill coefficient of the transition is ~3.0, indicating local cooperativity at the level of an E dimer and probably its immediate neighbors, but not an all-or-none transition over the entire particle surface.

Trimerization of soluble flavivirus E ectodomains (sE) at low pH generally requires the presence of liposomes, probably to accelerate subunit association when the fusion loops of monomeric sE insert into the lipid bilayer. Trimer formation under these conditions is irreversible. Trimerization of TBEV E on the virion surface is likewise irreversible, with an overall pH threshold of 6.5 (*Allison et al., 1995*). We measured irreversible trimerization of WNV sE (which is monomeric even at neutral pH), by following co-floatation with liposomes, and found a threshold pH of 6.1 (*Figure 5C*).

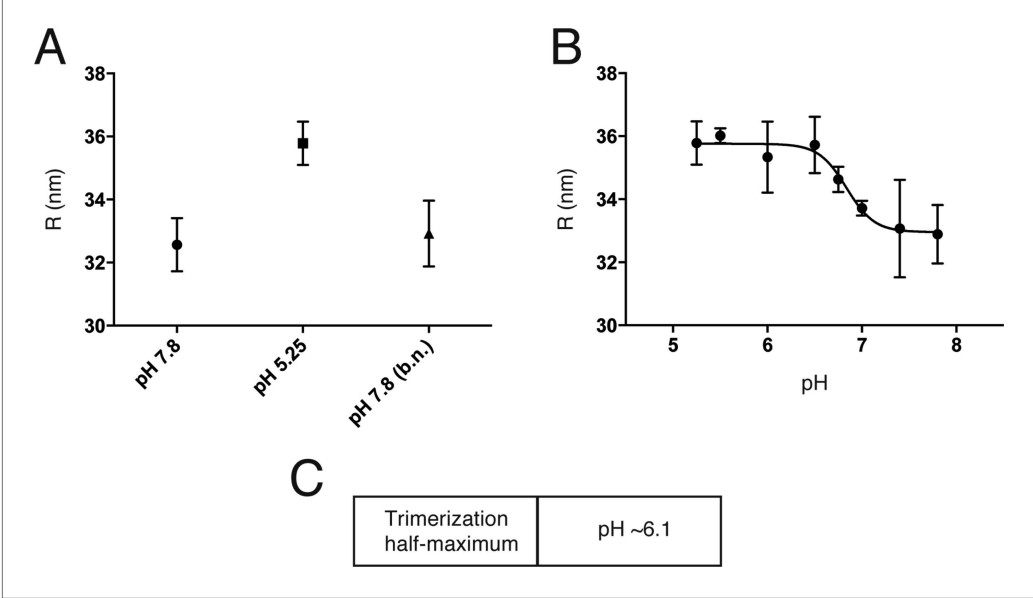

**Figure 5.** An initial reversible pH-dependent transition. (**A**) Reversible increase in hydrodynamic radius of Kunjin virus upon treatment with a low pH buffer (pH 5.25). The back-neutralized (to pH 7.8) sample shows the same initial radius. (**B**) Hydrodynamic radius measurements of Kunjin virus over a pH range 5.25–7.8. Half maximal extension observed at pH 6.8. The transition has a Hill coefficient of ~3.0. (**C**) pH half-maximal point of soluble, West Nile virus, E-protein trimerization as measured by liposome co-floatation.

## Simulations

Formal kinetic fits do not contain information about the underlying molecular mechanisms. We sought to relate the steps mediating hemifusion to molecular interactions between E-protein subunits on the VLP by carrying out a series of stochastic simulations for the activity of an array of E-proteins in contact with the supported membrane (*Figure 6A*). The design of the simulation came from our understanding of structural features of flavivirus virions, physical properties of the E protein, and direct measurements for different pH dependent transitions.

The contact patch in these simulations is a hexagonal array of 30 E monomers within a circular zone on the VLP (or virion) surface. The geometry is an idealization of the actual surface lattice, with explicitly defined dimers that reorganize during the dimer-to-trimer transition. We chose 30 monomers for the contact patch, because the small VLPs have about 60 subunits (depending on how perfect their surface lattice), and 30 subunits, a hemisphere, is the largest possible contact without major distortion of the target bilayer. The contact is unlikely to be much smaller, because particles with half of their E proteins containing 'weakened' fusion loops (the W101A mutation) had fusion properties similar to those of wild-type particles. A single number for the size of the contact patch is a reasonable approximation for the somewhat heterogeneous population of VLPs, because observed hemifusion kinetics did not depend on the extent of labeling with DiD and hence did not depend on the area of particle membrane or the diameter of the particle (*Figure 2F*).

We initialize the simulation by distributing the monomers between a 'prefusion' state and an 'activated' state. The former we take to be dimer-like; the latter we take to represent a conformation in which the monomer extends outward so that its fusion loops can contact the target membrane. The reversible transition between these states corresponds to the dimer-monomer equilibrium of soluble flavivirus E ectodomain (sE) and to the reversible E extension detected in our dynamic light scattering observations. It has a pH-dependent equilibrium constant, $K_{dm}$, and a forward rate constant, $k_{act}$ (and hence a probability $P_{act} = [1 - exp(-k_{act}\Delta t)]$ for each time step, $\Delta t$, in the simulation). Thus, $K_{dm} = (k_{act}/k_{ret})$ [H+], where $k_{ret}$ reflects the reverse rate constant (for the monomer to dimeric transition). We set the pK for $K_{dm}$ to agree with our measurements of the pH at half-maximal increase in the hydrodynamic radius (pH 6.8 for the ~30 Å expansion of Kunjin virus). We describe cooperativity in the activation transition with a factor ($P_{dim}$), by which the activation of one protomer increases the activation rate for its

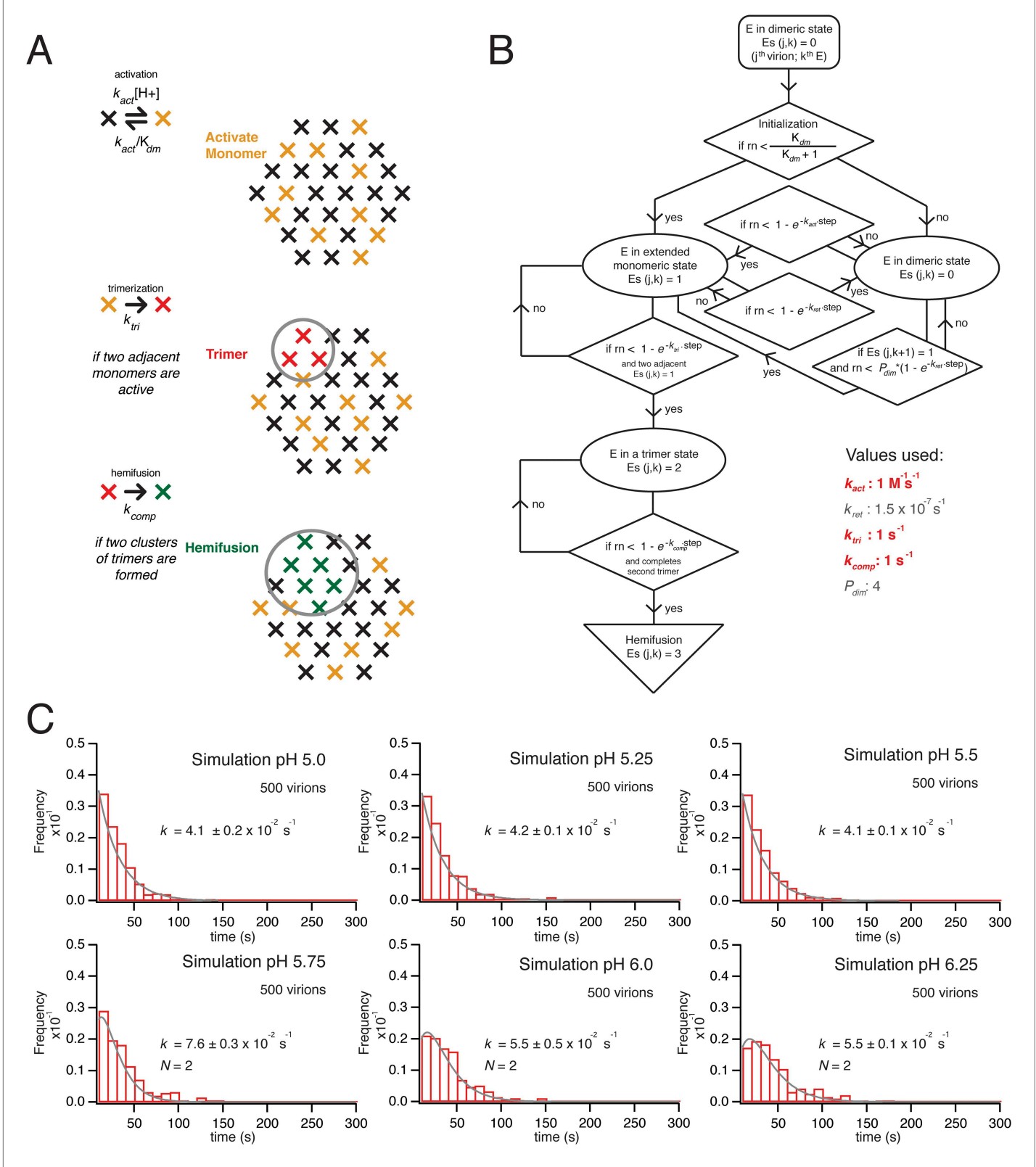

**Figure 6**. Simulation of time course to hemifusion. (**A**) Schematic for simulation of a flavivirus contact patch. E monomers are arranged in a hexagonal lattice (an idealization of the more complex arrangement on the virion surface). Each monomer can be in one of four different states: inactive monomer, active monomer, trimer member (with the condition that two adjacent monomers are active; if the condition holds, all three become trimer members),

*Figure 6. Continued on next page*

*Figure 6. Continued*

hemifusion mediator (with the conditions that it completes a trimer adjacent to another trimer). Dimers are defined explicitly as adjacent monomers in the lattice. The dimer-monomer transition is reversible; the trimerization and hemifusion steps are irreversible. The forward rate constant, $k_{act}$, reflects the dimer to monomer transition, while the reverse rate constant, $k_{ret}$, reflects the monomer to dimer transition. The pH-dependent equilibrium constant is $K_{dm} = (k_{act}/k_{ret})[H+]$. The assumption of reversibility in the dimer-monomer transition is based on dynamic light scattering observations (**Figure 5A**). Cooperativity of dimer activation is described by a factor, $P_{dim}$, by which the probability for dimer-to-monomer transition is multiplied. The rate $k_{tri}$ is the rate constant for completing one trimer. The rate constant $k_{comp}$ reflects the transition to hemifusion upon completion of the defined number of adjacent trimers. (**B**) Flow diagram of simulation, with final values optimized to the mean dwell time and fit over the experimental range of pH indicated in the figure. Free parameters shown in bold and red; fixed parameters, in gray. (**C**) Histograms for simulations of virions with $k_{act}$, $k_{tri}$ and $k_{comp}$ set to values shown in panel B, over the pH range 5.0–6.5. The curves for each histogram are a single exponential for pH 5.0–5.5 and a gamma distribution with $N = 2$ for pH 5.75–6.25.

neighboring dimer partner. That is, for a given neighbor, its probability for activation, $P_{act}$(neighbor) = $P_{dim}[1-\exp(-k_{act}\Delta t)]$, if (and only if) the partner is already activated (**Figure 6A**).

The extended monomers that bridge to the target membrane can then cluster as trimers. We define explicitly all possible trimer combinations for the 30 protomer hexagonal lattice. We assume that in this second transition, a trimer forms with probability $P_{tri} = [1-\exp(-k_{tri}\Delta t)]$, whenever three adjacent monomers (a triangle of positions in the idealized lattice of the simulations—**Figure 6A**) become activated; this step includes essentially irreversible capture of the fusion loops by the supported bilayer, as observed experimentally (**Stiasny and Heinz, 2004**). It corresponds to the observed irreversible trimerization and membrane capture of E ectodomains when the protein is exposed to reduced pH in the presence of liposomes. We used the measured pH for half-maximal trimerization (6.1), as determined by liposome co-floatation of recombinant soluble WNV E (**Figure 5C**), to set the pH dependence for the trimerization step.

Finally, when T adjacent trimers have formed, the simulation allows hemifusion to proceed to completion with a probability $P_{comp} = [1-\exp(-k_{comp}\Delta t)]$. A simulation for a given virion exits on execution of this step, after recording the total number of steps (i.e., the time to hemifusion) (**Figure 6B**).

We ran hemifusion simulations for 500 particles over a range of pH values. We found a unique set of parameters (**Figure 6B**) that generated histograms very similar in shape and in pH-dependence of dwell time to our experimental data from WNV VLPs (**Figure 5C**). The three values we varied were $k_{act}$, $k_{tri}$ and $k_{comp}$. We converged on these parameters following sequential rounds of optimization, testing fits for conditions in which one, two or three trimers defined hemifusion. We also tested a range of values for $P_{dim}$ from 2–50, thus altering the cooperativity of subunit activation, and values for T from 1 to 3. We found that the data fit best to models in which two trimers mediate hemifusion and that this number was insensitive to the assumed $pK_a$ for monomer activation and to the assumed threshold pH for trimer formation. We investigated if accounting for cooperativity between a subunit and its neighbors in the surface lattice other than its dimer partner would alter the outcome of our simulations. Incorporation of an additional factor did not make any changes in the properties of the simulation histograms and their fits. For all subsequent simulations, we fixed the parameter $P_{dim} = 4$, well within the tested range. With these parameters, the rate-limiting step is trimer formation. We noticed that as we increased the pH in these simulations, we lowered the yield of total particles reaching hemifusion during the fixed time frame of the simulation. We found the same trend in the experimental data (**Figure 7B**). In the simulations, the lower yield came from a reduced pool of activated monomer available to form trimers and a resulting geometric penalty for trimer possibilities. Relaxing this constraint, by increasing the target patch size to 37 monomers, increased overall event yield in the simulation; decreasing the target patch size to 23 decreased the yield (**Figure 7C**).

## Hemifusion kinetics of WNV mutants

We generated a series of mutant WNV VLPs, with mutations chosen to target different conformational states, in order to relate the sequential conformational transitions in E to changes in rate constants derived from simulation (**Table 1**). Mutation of key histidine residues identified to affect fusion eliminated all activity, while many mutations targeting the dimeric prefusion state had little effect on hemifusion kinetics or yield, consistent with previous observations (**Fritz et al., 2008**).

Residue F408 is part of a conserved contact that bridges the N-proximal segment of the 'zipped' stem of one subunit with domain II of the adjacent subunit in the postfusion trimer (**Klein et al., 2013**).

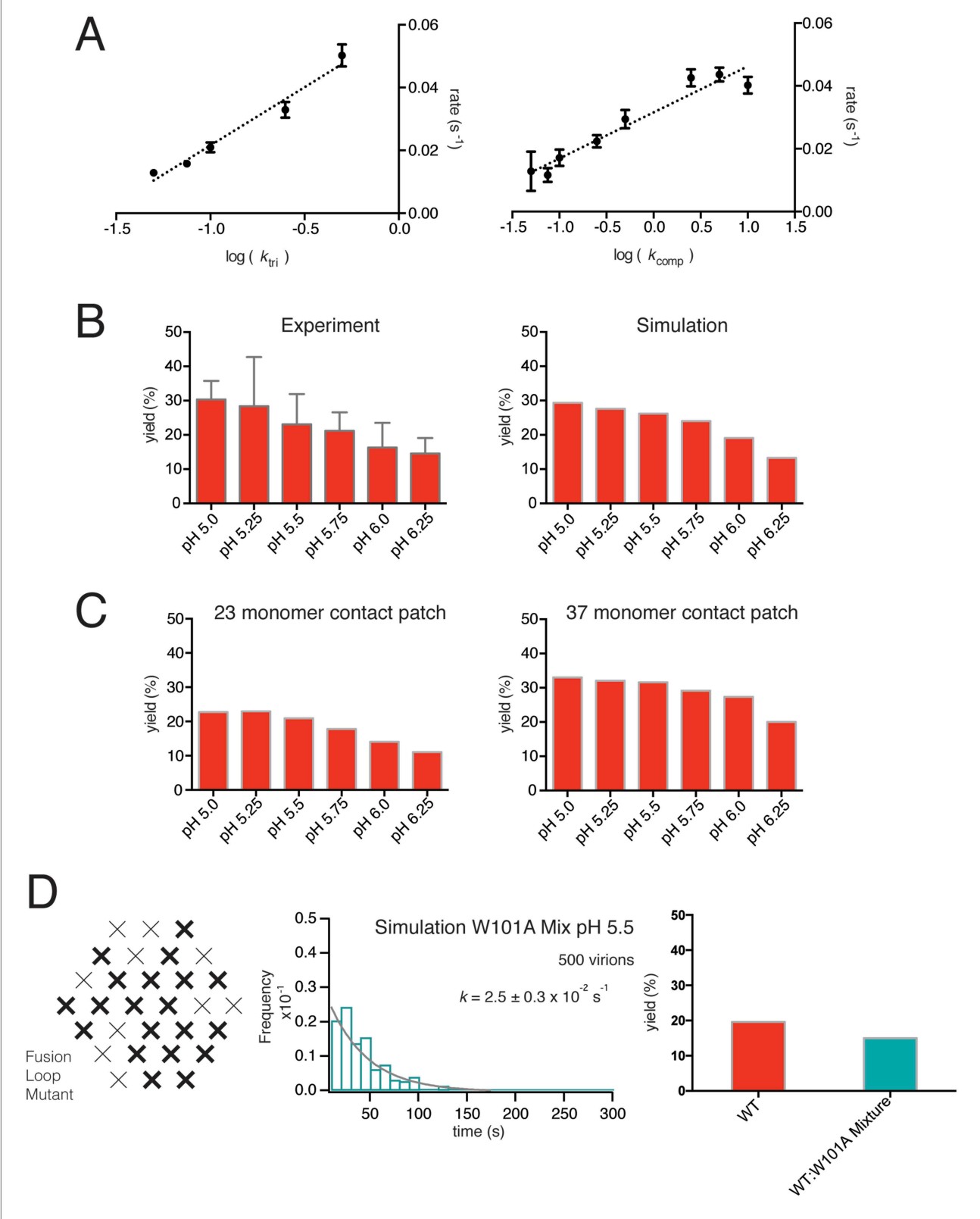

**Figure 7**. Simulations varying rate constants and contact-patch size and including mixture of wild-type and mutant E. (**A**) Overall hemifusion rate at pH 5.5 from simulations with all parameters fixed at values shown in *Figure 6B* except for $k_{tri}$ or $k_{comp}$ (left and right panels, respectively). (**B**) Yield of single particle hemifusion events (total number of fusion events/total number of identified particles in the field) as measured at different pH (left) and simulated

*Figure 7. Continued on next page*

*Figure 7. Continued*

with a 31-monomer contact patch (right), with $K_{dm}$ set at 6.8 and $k_{tri}$ set with a pH 6.1 half-maximal transition point. (**C**) Yield in simulations with 23-monomer and 37-monomer contact patches. (**D**) Simulation of time course and yield for mixed particles in which one-third of the E monomers could not stably engage the membrane but could be part of a trimer. One third was chosen for the proportion of dead subunits in simulation, because it is expected that the proportion of the mixed particles in the experiment with predominantly W101A subunits in their contact area would flow away upon pH drop and not be recorded. We assumed that trimers with one inactive monomer could participate in induction of a hemifusion stalk but that trimers with two or three inactive monomers could not. Compare with *Figure 4D*.

The interaction contributes to robust trimer formation in vitro of secreted, soluble TBEV E ectodomain (*Allison et al., 1999*). We measured the bulk fusion activity for F408V and found no variation in pH threshold, ruling out any global effect of the mutations on proton binding and restricting potential variation in the simulation parameters corresponding to $k_{tri}$ and $k_{comp}$ (*Figure 8A*). An F408V mutation in WNV VLPs slowed hemifusion at pH 5.5, broadening and extending the distribution of hemifusion delay times from a roughly single-exponential fall-off for wild-type at that pH (*Figure 3*, top right panel) to a rise-and-decay approximated reasonably by a two-step gamma distribution (*Figure 8A*). Under conditions expected to speed up the overall reaction (pH 5.0, 22°C), the F408V mutation reduced the single exponential rate for the distribution of dwell times from that of WT (*Figure 8B*). Simulations in which the trimerization rate constant ($k_{tri}$) for the F408V mutant was set to one-tenth its best value for the wild-type simulation generated a good fit to the observed hemifusion times, when all other parameters were held constant (*Figure 8C*). Simulations altering the $k_{comp}$ value generated poorer fits. The affected step in the simulation (trimerization) is consistent with the known contacts of F408 in the trimeric intermediate and with the known trimerization properties of E when those interactions are absent (*Pangerl et al., 2011*; *Klein et al., 2013*).

Residue N193, in the hinge region between domains I and II, contributes in the prefusion conformation to the 'bottom' of a cavity occupied by n-octyl-β-D-glucopyranoside (octyl glucoside) in a crystal structure of E from dengue virus serotype 2. In the post-fusion conformation, N193 makes a ring of hydrogen bonds in the center of the domain II trimer contact, where it probably contributes to clustering of the fusion loops. WNV N193A VLPs showed no change in bulk fusion threshold. Yet, single particle data of the WNV N193A mutant broadened the distribution of hemifusion delay times for the VLPs at pH 5.5, although somewhat less markedly than F408V (*Figure 8D*). Under conditions in which the reaction slows (pH 6.25, 18°C), the N193A data further broaden to a distribution of hemifusion times that fits a process with two steps of equal rate (*Figure 8E*). Reduction of $k_{tri}$ from its best fit to the wild-type distribution, produced only an approximate fit to this distribution (*Figure 8F*). Thus, the properties of the mutant are consistent with lower stability of the trimer relative to monomer and hence with an effect on late-stage trimerization of E, but it might also affect any other stage during which domain II rotation occurs.

Residues V434 and F450, both in the 'stem', are near the N-terminus of a membrane-embedded helix ('helix 2' in older terminology; 'α3' in the more recent assignment from fitting a high-resolution cryoEM map) and at the junction between stem and transmembrane anchor, respectively (*Figure 9A*). We collected single particle data for the V434W and F450W mutants, both of which showed no change in their bulk fusion threshold (*Figure 9B–C*). A single exponential best fit data from the F450W mutant, with a rate constant, $k = 1.2 ± 0.2 × 10^{-1}$. The F450W single particle data also had an extended tail in the dwell time distribution. The model from cryoEM reconstruction shows that F450 packs tightly into a hydrophobic cluster that includes a set of contacts with the N-proximal region of the M transmembrane anchor, close to the dimer twofold axis (*Zhang et al., 2013a*). Trimerization of E requires disruption of the packing around F450, to pull it away from M;

**Table 1.** Summary of mutant kinetic fits and rates

|  | $t_{mean}$ (s) | fit | $k$ (s) |
|---|---|---|---|
| WT | 46.1 | single exponential | $2.9 ± 0.4 × 10^{-2}$ |
| 1:1 WT W101A Mix | 53.4 | single exponential | $2.2 ± 0.4 × 10^{-2}$ |
| F408V | 95.8 | gamma N = 2 | $2.0 ± 0.2 × 10^{-2}$ |
| N193A | 85.3 | single exponential | $1.2 ± 0.2 × 10^{-2}$ |
| V434W | 65.5 | gamma N = 2 | $3.9 ± 0.3 × 10^{-2}$ |
| F450W | 54.2 | single exponential | $1.2 ± 0.2 × 10^{-1}$ |

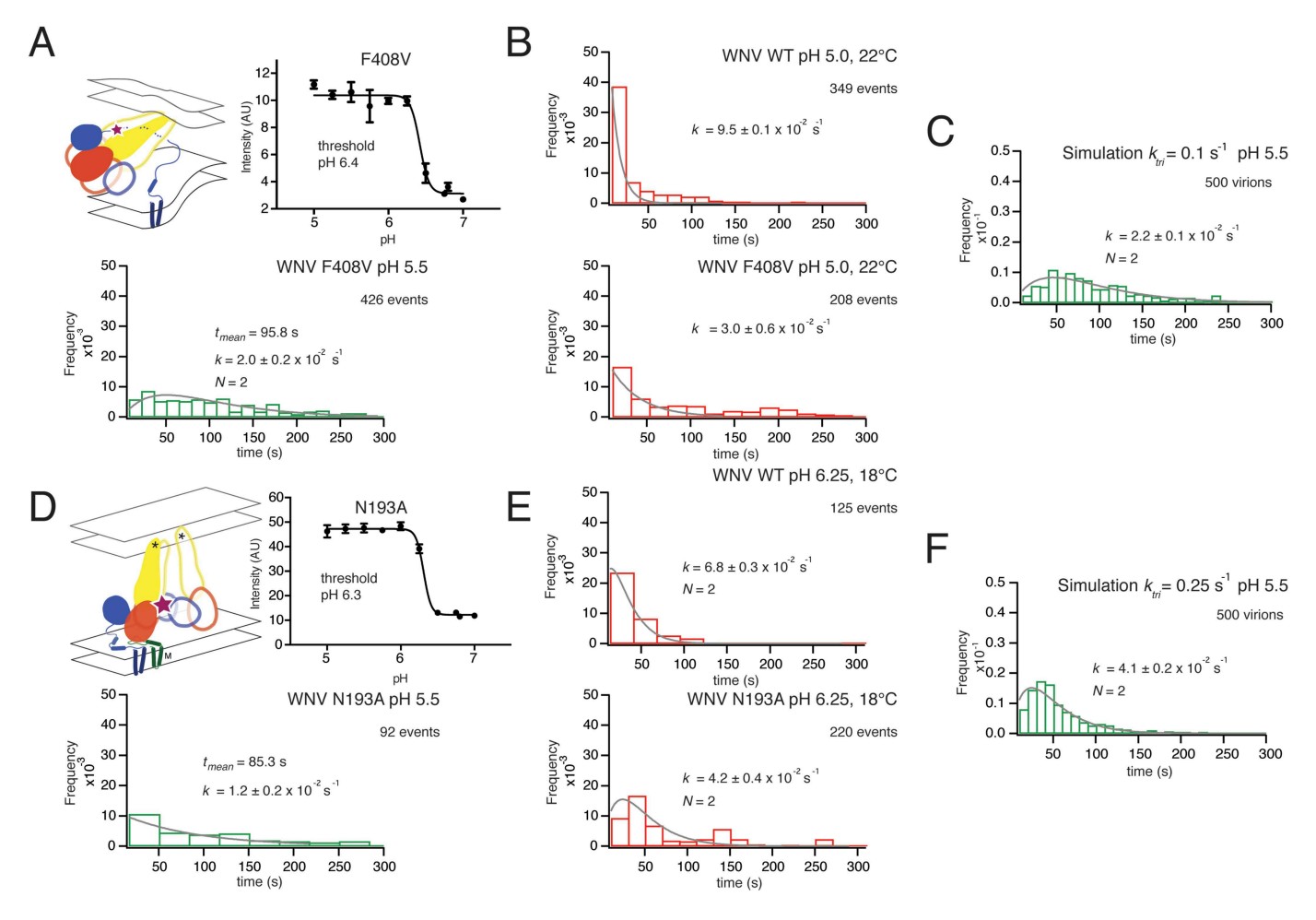

**Figure 8**. Effects of mutations in E on VLP hemifusion. (**A**) Schematic of conformational state targeted by F408V mutation (residue mutated indicated by star), bulk hemifusion titration and the single particle data histogram; compare with ***Figure 3A***, upper right-hand panel. (**B**) Single particle data for wild type and F408V WNV VLPs under conditions to accelerate the hemifusion reaction (pH 5.0, 22°C) (**C**) Simulation of WNV fusion with $k_{tri}$ parameter reduced to 0.1 s⁻¹. (**D**) Schematic of N193A conformational rearrangement (residue mutated indicated by star), bulk hemifusion data and single particle data; compare with ***Figure 3A***, upper right-hand panel. (**E**) Single particle data for wild type and N193A under conditions to further slow the reaction (pH 6.25, 18°C). (**F**) Simulation of WNV fusion with $k_{tri}$ parameter reduced to 0.25 s⁻¹.

subsequent zippering of the stem requires some perturbation of the way α3 interacts with lipid. We expect substitution of a bulkier residue for F450 to destabilize the local packing with M and therefore to lower the barrier to trimer formation; indeed, the observed rate constant for the mutant was substantially higher than the rate constant for wild-type particles (***Figure 3A***, top right-hand panel). The V434W mutant was most closely fit with a gamma distribution, N = 2, and a rate constant $k = 3.9 \pm 0.3 \times 10^{-2}$, faster than wild-type, but without further data we cannot assign structural correlates for the two steps.

## Discussion

Structures of viral fusion proteins, from x-ray crystallography and (more recently) from cryoEM, correspond either to an immature conformation, a mature, primed conformation, or a rearranged, postfusion conformation. The single-particle fusion experiments described here probe the transient, intervening states, inaccessible to direct structural analysis. By correlating fusion kinetics with specific, site-directed alterations in E, we have sought to determine the rate-limiting molecular events, the number of WNV E trimers needed to fuse, and the mechanism for coordinating conformational change among the several trimers that generate a single fusion pore.

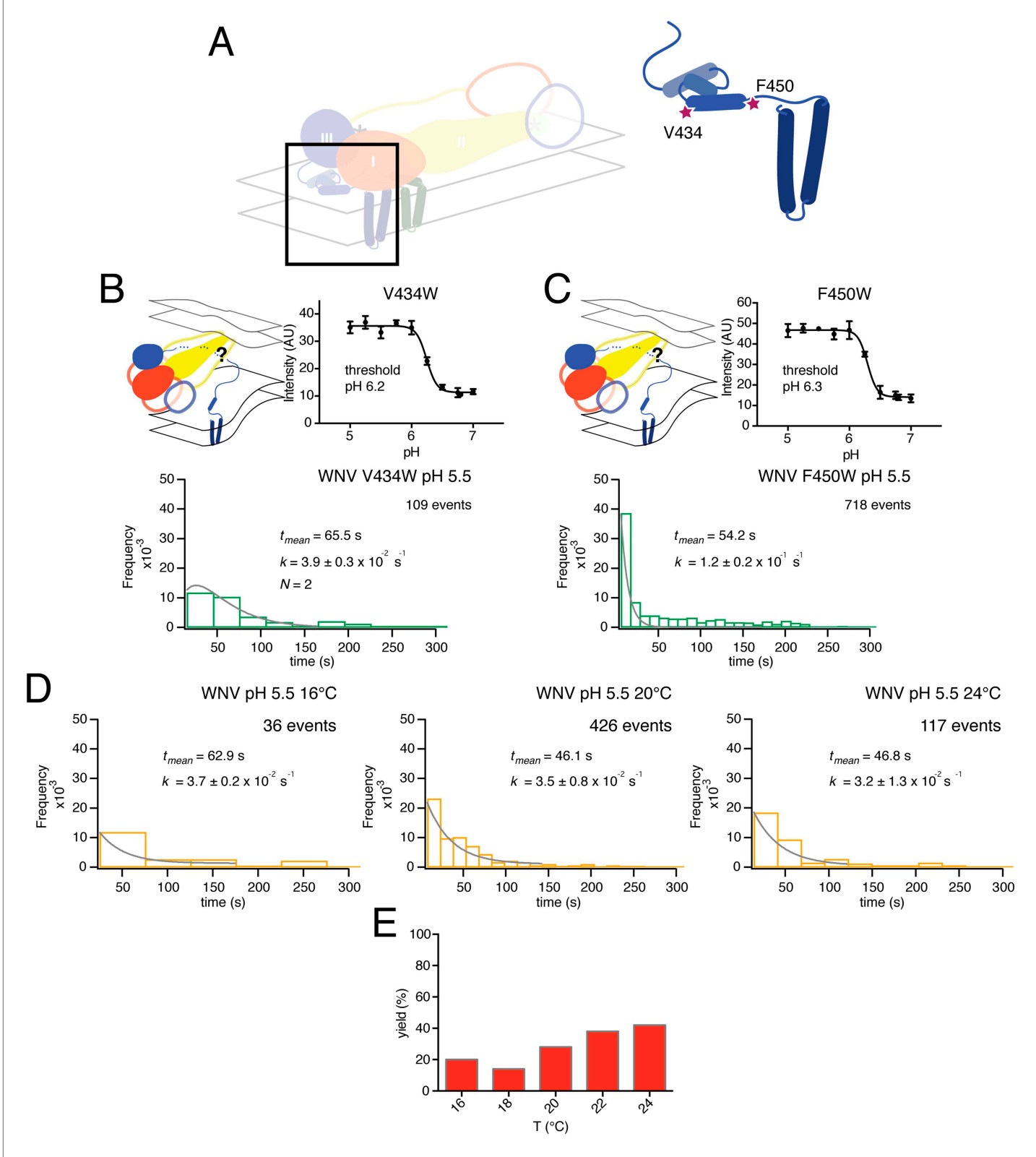

**Figure 9**. Effects of mutations in membrane-proximal segment of E and temperature on VLP hemifusion. (**A**) Schematic of stem region in pre-fusion state with mutations indicated by stars. (**B**) Schematic, bulk fusion data, and single particle data for V434W mutant at pH 5.5, 20°C; compare with *Figure 3A*, upper right-hand panel. (**C**) Schematic, bulk fusion data, and single particle data for F450W mutant; compare with *Figure 3A*, upper right-hand panel. (**D**) Single particle WNV VLP data at 22 and 24°C fit to a single exponential. (**E**) Yield of single particle events from 16 to 24°C.

Previous work on influenza virus, taking a similar approach, has produced the following description of hemagglutinin (HA) catalyzed fusion (*Floyd et al., 2008*; *Ivanovic et al., 2013*) (*Figure 9A*). The rate-limiting step in the hemagglutinin (HA) conformational change is exposure of the fusion peptide and its engagement with the target bilayer. The contact zone between a typical influenza virus particle and the membrane with which it will fuse includes about 100 closely packed HA trimers. When the pH drops, stochastic rearrangement of trimers within that contact zone creates a random pattern of extended HA bridges between the two membranes. The extended intermediates have a detectable lifetime only because hydration force and resistance to membrane deformation prevent collapse of any single trimer to a postfusion conformation. The free energy recovered from collapse of three neighboring trimers is enough to overcome this barrier, and rapid collapse and hemifusion ensue as soon as three adjacent trimers in the contact zone have created intermembrane bridges.

The sequence of molecular events in flavivirus fusion is somewhat more elaborate, because of the change in oligomeric association of E. E dimers, which initially form a regular icosahedral array, dissociate, exposing the fusion loops at the tip of domain II and allowing the subunits to project outwards from the virion surface. These events are fast, and under many conditions they may be in rapid equilibrium. Our results, both from experiment and simulation, suggest that the rate-limiting molecular step is formation of extended trimers. This conclusion is consistent with observations on low-pH induced trimer formation from soluble E ectodomain dimers: liposomes greatly enhance the yield of trimers, probably by transiently immobilizing the monomers in the right orientation. The rate of trimerization depends in turn on the effective surface concentration of activated monomers in the contact zone between virus particle and target membrane, and the rate of hemifusion, on the occurrence of a suitable number (at least two) of adjacent trimers within that zone. Since activation of individual monomers is stochastic, a critical constraint comes from the geometric requirements for availability of activated E to trimerize.

Previous, bulk-phase studies of flavivirus and alphavirus fusion with liposomes using pyrene-modified lipids incorporated into the virion followed the decrease in pyrene excimer fluorescence that accompanies dilution into the target liposome (*Chatterjee et al., 2000*; *Waarts et al., 2002*; *Fritz et al., 2011*). Our single particle analysis has extracted mechanistic detail from mutants that have no detectable bulk phenotype. The observed increase in single particle hemifusion yield at higher temperatures (~40% at 24°C) suggests that local rearrangements in the particle surface can help satisfy the geometric requirements for trimerization, increasing overall yield with little change in overall reaction rate (*Figure 9D–E*). Experimental data for the 1:1 WT:W101A mixed particle showed no decrease in hemifusion yield (*Figure 4D*). A likely explanation is that mixed trimers, with one or two Trp-containing fusion loops, can participate in facilitating hemifusion, perhaps in some cases with participation of a third trimer. A simulation assuming that any trimer with one or more mutated E monomers would be inactive greatly decreases yield, while relaxing this constraint recovers wild-type levels, as observed experimentally (*Figure 7D*).

The mechanism just outlined has some formal similarities to the one described for SNARE-catalyzed synaptic vesicle fusion, but the latter has an additional regulatory feature leading to a pause immediately before initial membrane mixing. Fast (ms) synaptic vesicular membrane fusion proceeds by rearrangement of multiple copies of the trans-membrane SNAP/SNARE complexes (*Takamori et al., 2006*; *Hernandez et al., 2014*), docked in a poised state analogous to the fusion-loop inserted, extended E-protein intermediate (*Figure 10B*). The synaptic fusion machinery may also act in a stochastic fashion, in which a minimum number of complexes are sufficient to catalyze fusion. Although subsequent events are very fast, the poised state precedes any membrane merger: the two bilayers remain distinct until triggering, and activation by calcium-dependent co-factors allows coordinated, rapid progression from the extended state to full pore formation, probably through a transient hemifusion stalk (*Diao et al., 2012*; *Lai et al., 2014*). The pre-assembly step in synaptic vesicle fusion thus avoids the penalty imposed on viral fusion by the requirement that a critical number of neighboring bridges accumulate *after* the triggering event (increased proton concentration) before they can zipper and collapse.

Synaptic vesicles and virus particles both have rather sharply bent lipid bilayers, with curvature imposed by the proteins that drive their budding from a cellular membrane. Moreover, both have high local protein density within the bilayer—at least 15% for flaviviruses and nearly 25% for synaptic vesicles (*Takamori et al., 2006*). We find that WNV VLP fusion kinetics are independent of particle size and hence of curvature within the relevant range, which spans a substantial difference in outer:inner area

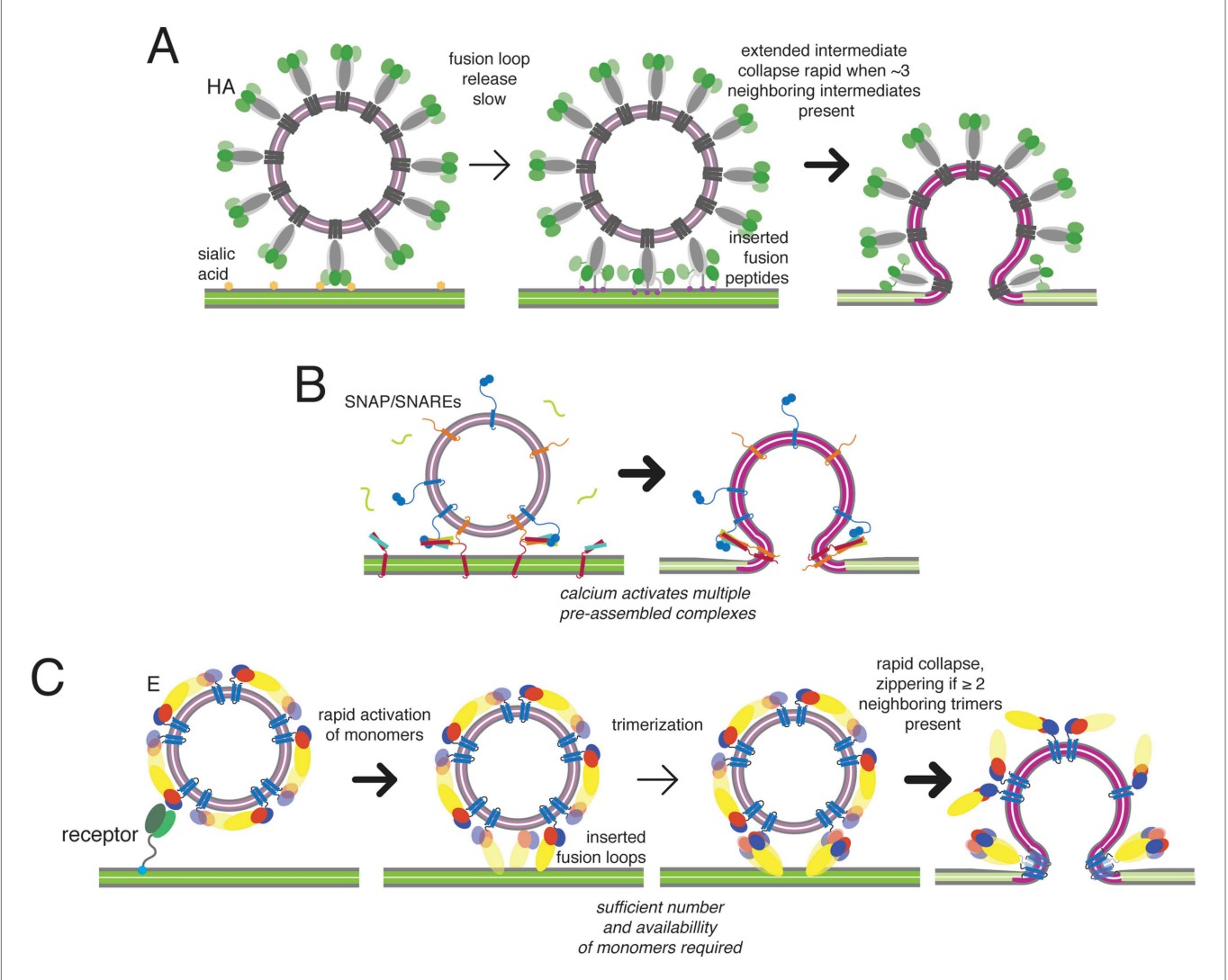

**Figure 10**. Kinetic mechanisms of membrane fusion. Darker arrows indicate faster steps. (**A**) Model of the sequence of events in influenza hemagglutin-mediated fusion. A representative schematic cross-section is shown (an actual contact zone will include 100 or more trimers). Stochastic release of a sufficient minimum number of fusion peptides is rate limiting. (**B**) Schematic of calcium-triggered SNAP/SNARE-mediated fusion. Rapid triggering and activation of a minimum number of assembled complexes by calcium. (**C**) Schematic of flavivirus fusion: monomer activation is fast (defined by pH), trimerization is rate limiting, and final collapse step is fast.

ratios. The area of the outer polar-group layer in small VLPs is nearly three times that of the inner polar-group layer; the ratio is just over 1.5 for the virion-sized particles. The wedge-shaped transmembrane hairpins of M and E, neither of which extends fully into the lumen of the particle, may compensate in part for curvature and help drive budding into the ER; insertion of amphipathic α3 of the stem into the outer leaflet of the bilayer, occupying about 10% of the total outer-leaflet surface area, probably stabilizes curvature as well. With these compensations, even the rather sharp curvature of the small VLPs probably contributes very little to overcoming the barrier to fusion. A potential contribution to catalyzing membrane merger is the perturbation in the target membrane that comes from introducing fusion loops or fusion peptides into its lipid bilayer. The heterogeneity of fusion-loop and fusion-peptide structures and sequences among the various kinds of enveloped viruses appears, however, to rule out a common mechanism for such a disturbance. Our analysis therefore concentrates on the distortions imposed by linking large-scale conformational changes in the fusion protein with comparably strong deformations of the lipid bilayers.

In the absence of regulation at a primed hemifusion step, like that regulated by synaptotagmin or complexin for SNAREs, the transition to fusion is very fast, once the necessary number of trimers has formed (*Figure 10B*). The requirement that several neighboring monomers undergo domain rearrangements to assemble into trimers implies that the extended conformation will have a finite lifetime and that blocking trimerization will markedly reduce the likelihood of hemifusion. The potency of exogenous stem-derived peptide inhibitors of dengue virus (*Schmidt et al., 2010*) suggests that these properties provide an inhibitory strategy yet to be fully exploited by conformation-specific targeted antibodies or small-molecule inhibitors.

## Materials and methods

### Virus-like particles (VLPs)

West Nile Virus virus-like particles (VLPs) were produced from a stable 293T cell line transfected with the pVRC8400 expression vector with a structural cassette containing prM-E sequence from the genotype NY99 sequence (produced by Angelica Medina–Selby, Doris Coit and Colin McCoin (*Lanciotti, 1999*)) preceded by the tissue plasminogen activator signal sequence. WNV VLPs were harvested at 37°C from Gibco FreeStyle 293 medium (Life Technologies, Grand Island, NY), clarified from debris by low-speed centrifugation and precipitated with Polyethylene glycol 8000. Following resuspension in buffer containing 20 mM Tricine (N-(2-Hydroxy-1,1-bis(hydroxymethyl)ethyl)glycine) pH 7.8, 140 mM NaCl and 0.005% Pluronic F-127, VLPs were purified over a Optiprep density gradient (SW41 rotor, 34,000 rpm, 4°C, 2 hr. 20 min.) with 55%-45%-35%-30%-25%-20–10% steps. We collected the band between the 35% and 30% densities and found this material to contain >95% fully processed M and to consist of particles 35 and 50 nm in diameter as assessed by cryo- and negative-stain electron microscopy (*Allison et al., 2003*). Particles were labeled with DiD (1,1′-Dioctadecyl-3,3,3′,3′-Tetramethylindodicarbocyanine Perchlorate) at ~20 μM or 20-fold the protein concentration. Excess dye was removed using NAP-10 desalting column (GE Healthcare, United Kingdom).

### Cell culture and viruses

C6/36 cells were maintained in L-15 medium (Mediatech, Manassas, VA) supplemented with 10% fetal bovine serum. For plaque-forming assays (PFAs), BHK-21 cells were in minimum essential medium (α-MEM) supplemented with penicillin, streptomycin, and 5% fetal bovine serum (FBS). Aliquots of Kunjin virus were purified by PEG precipitation and an Optiprep gradient and labeled at ~20 μM DiD (or with equivalent concentration DMSO). Ten-fold dilutions in EBSS were prepared in 48-well plates, and 100 μl of each dilution added to the cells. The plates were incubated for 1 hr at 37°C, unabsorbed virus removed by two washes with PBS, and 1 ml of α-MEM, supplemented with carboxymethyl cellulose (CMC), penicillin, streptomycin, and 2% FBS, was added to each well. After a 4-day incubation, the CMC overlay was removed, and the cells were washed with PBS and stained with crystal violet. The plates were washed with water to remove excess crystal violet and dried overnight.

### Flow-cell & supported bilayer preparation

Glass coverslips were cleaned by sonication in '7X' detergent, 1M potassium hydroxide, acetone and ethanol, and dried for 1 hr at 100°C. Polydimethylsiloxane (PDMS) flow cells with 0.5 mm wide and 70 μm high channels (5 per cell) were prepared as described previously (*Ivanovic et al., 2012*) and bonded to plasma-treated glass. Teflon FEP tubing (0.2 mm, Upchurch Scientific) connected an eppendorf tube with solution to the channel, and Intramedic polyethylene tubing (0.76 mm) connected the channel to a syringe pump (Harvard Pump 11; Harvard Apparatus, Holliston, MA).

Liposomes for preparing planar bilayers contained 1-palmitoyl-2-oleoyl-sn-glycero-3-phosphoethanolamine (POPE), 1-oleoyl-2-palmitoyl-sn-glycero-3-phosphocholine (POPC), cholesterol, and 1,2,-dioleoyl-sn-glycero-3-phosphocholine (DOPC), 1,2-dioleoyl-sn-glycero-3-phosphoethanolamine-N-(carboxyfluorescein) (FL-PE) and 1,2-dioleoyl-sn-glycero-3-[(N-(5-amino-1-carboxypentyl)iminodiacetic acid)succinyl] (Ni-NTA DOGS) (Avanti Polar Lipids, Alabaster, AL) in a ratio of 4:2:2:2:0.02:1%. Liposomes at 10 mg/ml were extruded through a 200 nm pore-size polycarbonate membrane filter. Liposomes were loaded into the flow cell and the flow then stopped to allow bilayers to form. We preformed fluorescence recovery after photobleaching experiments to confirmed the fluidity of the bilayer (*Figure 2D*). Unattached liposomes were washed away, and E16 Fab (or DCSIGN-R) with a C-terminal His6 tag was introduced at 50 nM for 2 min. His-tagged E16 was produced from a stable 293T line expressing both heavy and light chains from the pVRC8400 vector, purified by Ni-affinity chromatography

and S200 size-exclusion chromatography. DCSIGN-R and soluble WNV constructs was expressed from Hi-5 cells infected with recombinant baculovirus. Labeled virus particles were loaded onto pseudo-receptor decorated bilayer. To initiate fusion, we introduced acetate buffer (100 mM sodium acetate, pH 5.0–5.5) or MES (100 mM, pH 5.75–6.25), with 140 mM sodium chloride and 0.005% Pluronic F-127.

## Fluorescence

End-point bulk fusion data were collected using a GE Amersham Typhoon plate reader at 633 nm and 670 nm excitation and emission wavelengths respectively in 96-well clear-bottom plates with 2 mg/ml final lipid concentration (200 nm liposomes prepared as described above). VLPs were prepared and labeled with DiD as previously described.

Kinetic Bulk liposome fusion data were collected on a PTI (Photon technology International, Edison, NJ) 814 Fluorimeter at 648 nm and 669 nm excitation and emission wavelengths respectively. Data were collected with Cole–Parmer digital polyStat temperature controlled thermo-jacket at 2 Hz over 10 min and at 0.2 mM final lipid concentration (200 nm liposomes prepared as described above).

Single-particle fusion data were collected on an inverted Olympus IX71 fluorescence microscopy with a high numerical aperture objective (60×, N.A. = 1.3). VLPs were illuminated with 488 and 640-nm Coherent (Wilsonville, OR) lasers. A custom-fabricated water-chilled temperature collar (Bioptecs, Butler, PA) was fitted on the objective turret. Each time-lapsed fluorescence Video was recorded at 1 Hz for 300 s using 3i Slidebook software.

## Data analysis

The position of each particle was determined by particle tracking analysis. Fluorescence trajectories were calculated by integrating the intensities from a 4 × 4 pixel region around each particle. Data were analyzed with software written in MatLab (The Mathworks, Natick, MA), and Igor Pro (WaveMetrics, Lake Oswego, OR). The square-root of the number of observations was chosen for the number of bins when generating histograms.

As described in Floyd et al. (*Floyd et al., 2010*), a waiting time distribution contains information regarding the mechanism of the process in its shape. If a process proceeds through an intermediate, the waiting time distribution is the joint probability density, or the convolution of each individual process. If each transition is a single-exponential decay:

$$k_1 e^{[-k_1 \tau]}$$

then the Gamma distribution is the convolution of N exponential decays:

$$p(\tau) = \frac{k^N \tau^{N-1}}{\Gamma(N)} e^{-kt}$$

where $\Gamma(N) = (N-1)!$ for integral N.

For a process with two steps, the fit can be described (*Floyd et al., 2010*) by the equation:

$$p(\tau) = \begin{cases} \dfrac{k_1 k_2}{k_1 - k_2}\left(e^{-k_2 \tau} - e^{-k_1 \tau}\right), k_1 \neq k_2 \\ k^2 \tau e^{-k\tau}, \ k_1 = k_2 \end{cases}$$

For three steps, with three different rates:

$$p(\tau) = \frac{1}{(k_2 - k_1)(k_1 - k_3)(k_2 - k_3)} k_1 k_2 k_3 e^{-\tau(k_1 + k_2 + k_3)}$$

$$\left( (k_2 - k_1) e^{\tau(k_1 + k_2)} + (k_1 - k_3) e^{\tau(k_1 + k_3)} + (k_3 - k_2) e^{\tau(k_2 + k_3)} \right)$$

When two of the three rates are very similar, the distribution becomes:

$$p(\tau) = a^2 b \frac{e^{-b\tau} - e^{-a\tau} + (a - b)\tau e^{-a\tau}}{(a - b)^2}$$

where $k_1 = k_2 = a$, and $k_3 = b$.

We compared fits for models for a single exponential, a process with two rates (equal or not equal to one another), three rates (each independent), and three rates (with two set equal to one another).

In all data presented in this study, we used a simple Akaike information criterion test (AIC = $\chi^2 + 2k$, where $k$ = the number of parameters; it measures the relative quality of a statistical model by comparing the trade-off between goodness of fit and the number of parameters). Decisions on the model used for fitting were made on a case-by-case basis.

### Dynamic light scattering

Dynamic light scattering measurements were made at dilute concentrations (<0.5 mg/ml) to avoid aggregation and at 20°C using a DynaPro Protein Solutions instrument (Wyatt Technology, Santa Barbara, CA). Solvent refractive index and viscosity parameters were defined using the instrument PBS standard values. We calibrated the instrument using polystyrene beads (5 nm radius), tomato busy stunt virus in compact and expanded states (R = 170 Å and 190 Å, respectively), and rotavirus double-layered particle as standards (350 Å radius). Each data point is the average of three measurements, each of which was determined by twenty consecutive 10 s acquisitions Data were filtered based on stability of light intensity and sample polydispersity criteria (<20%) with the program DYNAMICS v6 for data analysis.

### Simulations

Simulations were written in MatLab (Mathworks). Code found in *Source code 1*.

## Acknowledgements

We thank Kevin Perkins and Jason Choi, for preliminary experiments; Dan Floyd and Irene Kim, for advice on assay development; Calixto Saenz and the HMS Microfluidics Facility, for help with instrumentation; Melissa Chambers, Junhua Pan, Yuhang Liu, Chen Xu and Niko Grigorieff, for help with electron microscopy; Eric Marino (Children's Hospital), for help with TIRF microscopy; Michael Diamond, for Kunjin virus; Xuling Zhu, Jenifer Kaplan, Jinhua Wang, Jaebong Jang, Priscilla Yang and Nathanael Gray, for discussions; Antoine van Oijen and Tijana Ivanovic, for comments on the manuscript. LHC is a Frederic M. Richards Fellow of the Jane Coffin Childs Memorial Fund for Medical Research. SCH is an investigator of the Howard Hughes Medical Institute. We acknowledge support from NIH grants CA13202, AI057159 (New England Regional Center of Excellence for Biodefense) and AI109740 (Center for Excellence in Translational Research).

## Additional information

### Competing interests

SCH: Reviewing editor, *eLife*. The other authors declare that no competing interests exist.

### Funding

| Funder | Grant reference number | Author |
|---|---|---|
| National Institutes of Health | CA13202, AI057159, AI109740 | Stephen C Harrison |
| Howard Hughes Medical Institute | | Stephen C Harrison |
| Jane Coffin Childs Memorial Fund for Medical Research | | Luke H Chao |

The funders had no role in study design, data collection and interpretation, or the decision to submit the work for publication.

### Author contributions

LHC, Conception and design, Acquisition of data, Analysis and interpretation of data, Drafting or revising the article; DEK, AGS, Conception and design, Drafting or revising the article; JMP, Acquisition of data, Drafting or revising the article; SCH, Conception and design, Analysis and interpretation of data, Drafting or revising the article

## Additional files

### Supplementary file

• Source code 1. Matlab code for simulations. File flavi_hemi.m contains the simulation for a flavivius E protein contact patch of 30 protomers.

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
