## [Decision Letter]

Thank you for sending your work entitled “Sequential conformational rearrangements in flavivirus membrane fusion” for consideration at *eLife*. Your article has been favorably evaluated by Randy Schekman (Senior editor), a Reviewing editor, and 3 reviewers.

The Reviewing editor and the reviewers discussed their comments before we reached this decision, and the Reviewing editor has assembled the following comments to help you prepare a revised submission.

Generally your work is considered to be novel and of high interest. However, the reviewers agree that the novelty lies primarily in the kinetic model which reveals, for instance, that trimerization is a rate-limiting step, rather than in the experimental approach or the general pathway involving dimer dissociation, trimerization etc. The modeling thus constitutes the key part of the work, and several aspects of it were viewed as rather problematic.

1) The simulation model has five fitting parameters (Figure 6). However, could there be other combinations of the parameters that would give similar quality of the fit? Can the combinations be different for different pH? For instance, can lipid mixing be mediated by several trimers at acidic pH and by only one at moderately acidic pH? In the analysis of the mutants four of the five parameters were kept fixed. Can a comparable fit be found by varying *k*_*comp*_? Also, are the simulations dependent on the grid size, i.e. the spacing of the trimers?

2) A related problem is that relevant experimental data are not well explained by the fit, such as why VLPs with 1-to-1 mix of wild-type E protein and W101A E protein had a mean dwell time and an overall yield of fusion events similar to those of fully wild-type particles. The yield of hemifused particles depends on pH in simulations much stronger than in the experiment (Figure 7).

3) Previous studies on several fusion processes suggested that there are higher energy barriers to overcome on the way from hemifusion to an expanding fusion pore than on the way from the tight contact to hemifusion. Should the conclusion “… makes trimerization a bottleneck in the overall course of the fusion reaction” be edited to “a bottleneck in hemifusion”?

In summary, the referees feel that more simulations with different parameters need to be carried out, and they also suggest the experimental data to be re-analyzed in order to obtain more data for fittings to additionally validate the model and conclusions (mostly yields as a function of pH for the mutants). It was felt that this should be doable within the time allowed for revision. Provided that by carrying out additional analysis and addressing the issues above the main conclusions are substantiated and robust, the reviewers are supportive of publication. However, it cannot be excluded that major conclusions need to be modified by the changes in the analysis, which may affect the novelty of the work.

[Editors’ note: further revisions were requested prior to acceptance, as described below.]

Thank you for resubmitting your work entitled “Sequential conformational rearrangements in flavivirus membrane fusion” for further consideration at *eLife*. Your revised article has been favorably evaluated by Randy Schekman (Senior editor) and a member of the Board of reviewing editors. The manuscript has been improved but there are some remaining questions concerning the modeling that should be addressed before acceptance.

For one of the two fixed parameters –*P*_*dim*_, the authors now tested a range of values and found the data still fit best to models with two trimers mediating hemifusion. This is an important analysis and it would be useful to specify the tested range. The second fixed parameter is the pK for K_dm_. In the Rebuttal letter the authors wrote: “we have used experimental data on the conformational transition of soluble E to fix the pK of the dimer-monomer step”. This would be great, but the description in the manuscript is actually very different. The manuscript states: "we set the pK for K_dm_ to agree with bulk measurements of the pH threshold for fusion (pK_f_, pH 6.4 for WNV VLPs). This has to be clarified.

If the pK of the dimer-monomer step is indeed directly measured, please describe these measurements in the paper.

If pK for the dissociation of the WNV E dimer is fixed at 6.4 based on the literature, please provide the references. I do not think References 1 and 20 mentioned in the Rebuttal (“a well-characterized and biochemically established condition is the pH threshold for dimer dissociation (References 1, 20)”) help to set this pK, since Reference 1 is a study carried out on another virus (tick borne encephalitis rather than WNV) and Reference 20 does not present any data for the pH threshold of the dimer dissociation (only pH dependence of lipid mixing).

If the pK of the dimer-monomer step is taken from the pH dependence of lipid mixing, this is not a direct measurement. Dimer dissociation is not the only low pH-dependent stage in fusion, and other pH-dependent stages (including the trimerization step that is found by the authors to depend on pH with pH 6.1 threshold) can contribute to or determine the observed pH dependence of fusion. Furthermore, for the same viral fusogen, pH threshold for lipid mixing can depend on the specifics of the assay (for instance, Wessels et al., 2007, Biophys. J., 93, 526; [23], Journal of General Virology, 91, 389) and on the composition of the target membrane (for instance, Gollins & Porterfield, 1986, Journal of General Virology. 67, 157). Thus, pK of the dimer dissociation step can differ from pH threshold of lipid mixing, and it is important to examine whether small changes in the pK for K_dm_ would yield better fits with 1 or 3 trimers mediating hemifusion or 2 trimers would remain the best fit.

---

## [Author Response]

*Generally your work is considered to be novel and of high interest. However, the reviewers agree that the novelty lies primarily in the kinetic model which reveals, for instance, that trimerization is a rate-limiting step, rather than in the experimental approach or the general pathway involving dimer dissociation, trimerization etc. The modeling thus constitutes the key part of the work, and several aspects of it were viewed as rather problematic*.

The principal concerns were whether there are too many free parameters in the simulation and whether we could improve the fit with yield for mixed particles of W101A (“dead” fusion-loop mutant). We now point out explicitly that of the five parameters, only three are free, because we have used experimental data on the conformational transition of soluble E to fix the pK of the dimer-monomer step and because we fixed the cooperativity parameter at *P*_*dim*_=4. We provide details below. We have also found that by introducing a pH dependence for the trimerization step, again fixed by experimental measurement, and by allowing trimers with one “dead” protomer to participate in hemifusion, we considerably improve the agreement of simulated and observed yield. Again, further details below.

*1) The simulation model has five fitting parameters (*Figure 6*). However, could there be other combinations of the parameters that would give similar quality of the fit? Can the combinations be different for different pH? For instance, can lipid mixing be mediated by several trimers at acidic pH and by only one at moderately acidic pH? In the analysis of the mutants four of the five parameters were kept fixed. Can a comparable fit be found by varying* k_comp_*? Also, are the simulations dependent on the grid size, i.e. the spacing of the trimers*?

We emphasize that our model in fact has only three free parameters. A well-characterized and biochemically established condition is the pH threshold for dimer dissociation. This value determines the K_dm_ equilibrium constant in our simulation, thus fixed the ratio of *k*_*act*_ to *k*_*ret*_. We fixed *P*_*dim*_ (the cooperativity factor for dimer dissociation) at 4, corresponding roughly to a four-fold increase in the dimer to monomer transition rate. Thus, the three values we varied were those of *k*_*act*_, *k*_*tri*_ and *k*_*comp*_. We found that other combinations could not generate as good a fit. The text and figures have been modified to make this clear.

We considered whether models with a different number of trimers required to complete hemifusion could fit our experimental data equally well. We compared the results of simulations in which we varied the number of trimers defining hemifusion (Figure 11). The model with two trimers clearly fit better than models with one or three trimers. This global behavior applies across the pH range studied. Attempts to optimize the fitted parameters with a three-trimer or one-trimer hemifusion model resulted in poor fits of the simulation distributions to the experimental data.Author response image 1.

We also investigated whether we could generate better agreement with the data from mutants by varying *k*_*comp*_, as suggested, and found that the fits using this model were poorer (as reflected by the χ^2^ value and plotted residuals), despite adding a second fitting parameter in the *k*_*comp*_ fit (Figure 12).Author response image 2.

Grid spacing was not a consideration for describing flavivirus fusion, as unlike influenza hemagglutinin, there is a regular icosahedral arrangement of E on the virion surface. We did note potential differences due to grid area (total number of E proteins in the contact zone) as observed in simulations (Figure 6).

*2) A related problem is that relevant experimental data are not well explained by the fit, such as why VLPs with 1-to-1 mix of wild-type E protein and W101A E protein had a mean dwell time and an overall yield of fusion events similar to those of fully wild-type particles. The yield of hemifused particles depends on pH in simulations much stronger than in the experiment (*Figure 7*)*.

The difference between our experimental observations and the simulation results for the mixed particles prompted us to consider whether a trimer containing two wild-type fusion loops might still be competent to mediate hemifusion. We generated another version of our simulation scheme to account for this possibility and found an overall event yield similar to wild type values, better matching our experimental observations (Updated Figure 7). The figures in the main text have been updated to reflect the new simulation.

We had considered that pH might influence different stages of the reaction, and in the originally submitted version of the work, we chose to build in a pH-dependence for the initial dimer dissociation step. To determine whether additional steps might depend on pH, without adding further free parameters, we measured the trimerization pH threshold for soluble WNV E protein in a liposome co-floatation assay and obtained a value of 6.1. Because WNV E is monomeric in solution, the measurement deconvolves dimer dissociation from trimerization. We built this parameter into our simulation, just as we had incorporated pH dependence with a measured pK into the dimer dissociation step and found better correlation between our simulation and the experimental yield at higher pH values (updated Figure 7).

*3) Previous studies on several fusion processes suggested that there are higher energy barriers to overcome on the way from hemifusion to an expanding fusion pore than on the way from the tight contact to hemifusion. Should the conclusion “… makes trimerization a bottleneck in the overall course of the fusion reaction” be edited to “a bottleneck in hemifusion”*?

We agree, and we have changed the wording as suggested.

[Editors’ note: further revisions were requested prior to acceptance, as described below.]

The reviewers’ principal concern was with the assumption that the pH threshold for bulk fusion is a suitable estimate for the pK of the dimer-monomer step on the virion surface. The point is a good one, and we realized that we could fix the parameter for K_dm_ (the pH-dependent dimer-monomer equilibrium) by measuring the dimer-monomer equilibrium step directly by dynamic light scattering of Kunjin virions. We have done so and incorporated this experimental value as a fixed parameter in the simulations. We now describe this measurement in the paper.

*If the pK of the dimer-monomer step is indeed directly measured, please describe these measurements in the paper*.

We used Kunjin virus, the West Nile virus variant; its uniform diameter allows us to interpret a change in dynamic light scattering as a change in effective hydrodynamic radius. We observe a reversible expansion upon low-pH treatment (new Figure 5). West Nile virions have been shown to undergo transient temperature dependent fluctuations, during which extension of the E protein opens access to otherwise sequestered epitopes (7). A similar transition seen with dengue virions is temperature dependent and can be trapped by binding of antibody to the epitopes it exposes (35; 36). The measured radius is somewhat larger than the fully compact structure seen by cryoEM, suggesting that E subunits can undergo transient, reversible outward excursion even at pH ∼8 and room temperature (see main text for further discussion).

We find that pH 6.8 is the midpoint of the transition from a smaller to a larger radius (new Figure 5). We interpret the change in hydrodynamic radius as a measure of the rearrangement of E subunits from the circumferential orientation of their long axis seen in mature virions to a partially extended conformation with fusion loops exposed. We take the midpoint pH for this transition as the pK of the dimer-monomer equilibrium for E protein on the virion surface (Figure 5). We have re-run all of the simulations in the paper with the newly determined K_dm_ value and updated all of the relevant figures (Figures 6, 7 and 8).

This transition shows a Hill coefficient of ∼3, indicating local cooperativity for the transition, but not a globally cooperative reorganization of the entire surface lattice.

*For one of the two fixed parameters –*P_dim_*, the authors now tested a range of values and found the data still fit best to models with two trimers mediating hemifusion. This is an important analysis and it would be useful to specify the tested range*.

We tested the *P*_*dim*_ parameter over a range from 2 to 50, as now included in the updated main text. In addition, based on the observed Hill coefficient of 2.5, we investigated if accounting for additional cooperativity effects on other adjacent subunits outside of a dimeric unit would alter our simulations. Incorporation of an additional factor did not make any changes in the properties of the simulation histograms and their fits. This result suggests that we can use a single *P*_*dim*_ factor to introduce cooperativity into the model, and that this factor is an adequate surrogate for the cooperative dimer to monomer transition on the surface of the virion.

*(…) Thus, pK of the dimer dissociation step can differ from pH threshold of lipid mixing, and it is important to examine whether small changes in the pK for K*_*dm*_
*would yield better fits with 1 or 3 trimers mediating hemifusion or 2 trimers would remain the best fit*.

We once more examined whether our simulation, with the pK value of 6.8 determined by dynamic light scattering, alters the agreement of models with 1 or 3 trimers mediating hemifusion and found that a model with two trimers remains the best fit (Figure 13).Author response image 3.